# Cardiovascular Magnetic Resonance from Fetal to Adult Life—Indications and Challenges: A State-of-the-Art Review

**DOI:** 10.3390/children10050763

**Published:** 2023-04-23

**Authors:** Sara Moscatelli, Isabella Leo, Veronica Lisignoli, Siobhan Boyle, Chiara Bucciarelli-Ducci, Aurelio Secinaro, Claudia Montanaro

**Affiliations:** 1Inherited Cardiovascular Diseases, Great Ormond Street, Children NHS Foundation Trust, London WC1N 3JH, UK; 2Paediatric Cardiology Department, Royal Brompton and Harefield Hospitals, Guy’s and St Thomas’ NHS Foundation Trust, London SW3 5NP, UK; 3Department of Experimental and Clinical Medicine, Magna Graecia University, 88100 Catanzaro, Italy; 4CMR Unit, Cardiology Department, Royal Brompton and Harefield Hospitals, Guy’s and St Thomas’ NHS Foundation Trust, London SW3 5NP, UK; 5Department of Cardiac Surgery, Cardiology, Heart and Lung Transplantation, Bambino Gesù Children’s Hospital IRCCS, 00165 Rome, Italy; 6Adult Congenital Heart Disease Department, Royal Brompton and Harefield Hospitals, Guy’s and St Thomas’ NHS Foundation Trust, London SW3 5NP, UK; 7Cardiology Department, Logan Hospital, Loganlea Rd, Meadowbrook, QLD 4131, Australia; 8School of Biomedical Engineering and Imaging Sciences, Faculty of Life Sciences and Medicine, King’s College University, London SW7 2BX, UK; 9Radiology Department, Bambino Gesù Children’s Hospital IRCCS, 00165 Rome, Italy; 10National Heart and Lung Institute, Imperial Collage London, Dovehouse St, London SW3 6LY, UK

**Keywords:** fetal CMR, CMR, pediatric CMR, cardiovascular magnetic resonance

## Abstract

Cardiovascular magnetic resonance (CMR) imaging offers a comprehensive, non-invasive, and radiation-free imaging modality, which provides a highly accurate and reproducible assessment of cardiac morphology and functions across a wide spectrum of cardiac conditions spanning from fetal to adult life. It minimises risks to the patient, particularly the risks associated with exposure to ionising radiation and the risk of complications from more invasive haemodynamic assessments. CMR utilises high spatial resolution and provides a detailed assessment of intracardiac and extracardiac anatomy, ventricular and valvular function, and flow haemodynamic and tissue characterisation, which aid in the diagnosis, and, hence, with the management of patients with cardiac disease. This article aims to discuss the role of CMR and the indications for its use throughout the different stages of life, from fetal to adult life.

## 1. Introduction

Cardiovascular magnetic resonance (CMR) imaging offers a non-invasive imaging approach which provides a highly accurate and reproducible assessment of cardiac morphology and function for use in the diagnosis and management across a wide spectrum of cardiac conditions spanning from fetal to adult life [1].

CMR is a useful diagnostic tool in the antenatal diagnosis and assessment of cardiac conditions, demonstrating good correlation with post-natal neonatal findings and has an important role in assessing response to in utero cardiac surgery. In all types of congenital heart disease in children, and in acquired forms of cardiac disease in the paediatric population, CMR can provide a thorough and tailored anatomical and physiological assessment utilizing specific targeted sequences and scanning protocols. CMR assessment may include evaluation of cardiac and extra-cardiac anatomy, vascular and valvular flow haemodynamic, shunt quantification to guide therapeutic management, and assessment of myocardial function. Functional assessment can be measured accurately by CMR in childhood with high reproducibility and avoidance of ionising radiation, regardless of ventricular morphology [2,3].

CMR is not without its challenges. There are unique challenges in each of the populations discussed. Examinations may be time consuming and require management of arrhythmias and breathing, which can negatively impact the image quality [1]. There are often additional challenges involved in the scanning of paediatric and neonatal patients where extra preparation and scanning time must be factored in and general anesthesia is often required for the scan to be carried out safely and effectively [2]. There are further considerations when performing CMR during pregnancy such as whether the scan can be carried out to the first trimester and without the use of gadolinium-based contrast agents (GBCA). Another consideration is that CMR in many centres is often a resource with limited availability and is more expensive compared with alternative cardiac imaging [4].

In adults, serial CMR is preferred to serial CT in patients with congenital heart disease or aortopathies due to the absence of ionising radiation. The combined and accurate anatomical and haemodynamic assessment in adults with congenital heart disease has established CMR as a valuable tool in managing patients with adult congenital heart disease [5]. CMR has a central role in the investigation of acute and chronic chest pain where microvascular obstruction can be identified, myocardial viability can be established, and tissue characterisation assists in the diagnosis or exclusion of pericardial disease and can help differentiate between an infarct pattern and myocardial inflammation or fibrosis from other causes. CMR has an important role in identifying the aetiology and performing the follow up on cardiomyopathies and cardiac masses [6].

Advances and technical improvements in CMR have facilitated the use of quantitative stress CMR for the evaluation of ischaemia and microvascular dysfunction as a non-invasive and comparatively accurate approach when compared with invasively derived parameters [7].

Cardiac magnetic resonance imaging (MRI) has a high spatial resolution and offers tissue characterisation via a non-invasive approach. It has the benefit of high-resolution images without requiring ionising radiation and is not reliant on an optimal acoustic window for a high-quality image. Cardiac MRI is useful as a sole modality in the diagnosis of many cardiac conditions but offers an additive benefit in combination with other more frequently utilised and more easily accessible imaging modalities. This article aims to discuss the role of CMR and the indications for its use throughout the different stages of life [1].

## 2. Role of Cardiovascular Magnetic Resonance during Fetal Life

### 2.1. Background

Congenital heart diseases (CHDs) are the most common lesions at birth affecting approximately eight in one thousand newborns [8]. These defects have a broad variety of severity, requiring, in a few cases, immediate postnatal interventions to improve survival [9]. The development of fetal echocardiography has led to the possibility of promptly organising postnatal interventions to reduce morbidity and mortality of the underline conditions [9]. In addition, fetal echocardiography has allowed families to make a decision regarding the pregnancy’s continuation in case of severe heart defects and to intervene during fetal life to improve the prognosis in case of approachable diseases [10,11]. Despite the significant advantages, fetal echocardiography has limitations that reduce the clinical fetal assessment. The echo data depends on the quality of the acoustic window, which tends to deteriorate at the end of pregnancy due to multiple factors such as maternal obesity, oligohydramnios, adverse fetal positions, multiple gestations, and acoustic shadowing by the bony thorax. Operator experience is also important. Fetal cardiovascular magnetic resonance (CMR), an imaging method not based on ionising radiation, can overcome acoustic window limitations and operator-dependent data evaluation, bringing the necessary information for the best management and outcome [12,13].

### 2.2. Overview of the Principal Sequences Used during Fetal CMR Scans

A fetal CMR study comprises different sequences that give information regarding anatomy, function, haemodynamics [13]. Balanced steady-state free precession (bSSFP) or “bright blood” sequences and single-shot fast spin echo or “black blood” sequences are used to comprehend the cardiovascular’ system anatomy [13]; in fact, thanks to their high spatial resolution and short time of acquisition, these can image a small moving heart. In postnatal CMRs, image acquisition is triggered by the electrocardiographic (ECG) signal that allows us to acquire information on the myocardium in the different phases of the cardiac cycle [14]. Unfortunately, this method cannot be applied in fetal life due to the absence of a direct fetal ECG recording, whereas other methods such as metric optimised gating (MOG), strategies based on “self-gating” or Doppler Ultrasound gating (DUS) tend to be used [15,16,17,18]. Through these techniques, balanced steady-state free precession (bSSFP) sequences can be acquired without losing ventricular contraction information, but artefacts might be present due to fetal movements. However, faster k-space sampling techniques can speed the acquisition, overcoming this inconvenience [19].

In both paediatric and adult CMRs, flow measurements are obtained through sequences named velocity-encoded cine phase-contrast (PC) that are ECG triggered as well. Consequently, flow quantification is triggered with other systems similar to the ones described above [11,20,21].

Finally, the magnetic resonance imaging (MRI) signal is sensitive to oxygen saturation and haemoglobin concentration changes. T1 and T2 mapping are sequences able to quantify the oxygen content of blood in fetal vessels providing an entirely novel approach to assessing fetal hemodynamics [22,23].

### 2.3. Fetal CMR Applications in the Clinical Scenario

CMR sequences available for fetal studies allow the detection of cardiovascular anatomy, function, and hemodynamics data. Studies have indicated good agreement between fetal CMR and echocardiography information and between CMR and post-natal findings. In addition, other research has shown a good concordance between the echocardiographic and CMR measurements [24,25,26]. Indeed, fetal CMR data seems helpful in adjunct to echocardiography for diagnosing CHDs, adding precious information for managing these patients [27]. The cine stacks quality and spatial resolution of bSSFP have been reported to be superior to the cine sweeps done by echocardiography [28]. Several studies have focused on the diagnostic value of fetal CMR in vascular anomalies with promising results [29]. For example, the transverse aortic arch view is relatively easy to obtain, reproduce, and interpret, helping with the challenging diagnosis of coarctation during fetal life [29].

In the past 15 years, different technical strategies have been proposed to overcome the lack of conventional cardiac gating methods and to reduce heart motion artefacts improving image quality (i.e., metric optimized gating, motion-corrected slice-volume registration) [15,30] and producing reliable CMR measurements other than additional anatomical information. More recently, a Doppler ultrasound gating system has been introduced in the market to synchronize data acquisition with fetal heart movement. The combination of cine PC flow sequence data and oximetry measurements based of blood pool signal of parametric imaging (T1 and T2 mapping) showed a more complete picture of fetal hemodynamics and general outcome not only related to the underlying heart condition [20]. This sequence combination can demonstrate blood flow and oxygen redistribution in CHDs. It has been seen that the obstruction of blood flow across one side of the heart increases the output of the contralateral ventricle. Surprisingly, this has not been demonstrated in single ventricle physiology, where the combined ventricular output is reduced [31]. This reduction is even more pronounced in severe atrioventricular valve regurgitation, showing reduced pulmonary, cerebral and umbilical perfusion [32]. The reduced cerebral perfusion may contribute to impaired brain growth and development, which has been seen in newborns with CHDs [32].

The demonstration of reduced fetal oxygen delivery and cerebral hypoxemia in fetuses with CHDs has led to an interest in the use of maternal hyperoxygenation as a neuroprotective strategy [32]. A pilot safety and feasibility study of chronic maternal hyperoxygenation in fetuses with single ventricle CHD (NCT03136835) is now in process to see if any improvement is possible.

Fetal CMR can aid decision making around fetal cardiac intervention and monitor the impact of an in utero procedure, such as for utero atrio-septostomy [33].

### 2.4. Current Limitations

Fetal CMR is an emerging technique that is improving the managing of CHDs in the pre-natal era. However, it has some limitations that require future studies and research. It is still limited to the third trimester, and despite the great postprocessing improvement, the moving artefacts sometimes remain challenging to overcome. The investigation and post-processing are still time-consuming and difficult to have. In addition, the method is expensive, and only specialised centres can assess it.

Finally, intravenous gadolinium is teratogenic in animals at high and repeated doses because it crosses the placenta and is excreted by the fetal kidneys into the amniotic fluid. Thus far, no concerns have been reported in a human fetus. The current indication is to use gadolinium when the benefits outgrowth the potential risks, although most clinicians are inclined to avoid its administration [4].

## 3. Role of Cardiovascular Magnetic Resonance during Childhood

### 3.1. Introduction

The survival of CHDs patients has also increased because of improvements in early diagnosis and treatment, which have led to more patients surviving into adulthood [34]. However, long-term morbidity and mortality are substantial, as is the need for reinterventions [35]. Furthermore, there is an increasing number of children with acquired heart disease, particularly chemotherapy cardiotoxicity, following treatment of oncological disease in early childhood.

Imaging is fundamental to the diagnosis of CHDs and is required at all stages of patient care. As previously mentioned, from the fetal stage onwards, imaging outlines anatomy and physiology, helps to refine management, evaluates the consequences of interventions, and helps guide the prognosis. However, no single available imaging modality fulfils these roles for all patients and diseases. Specific challenges apply to imaging children and adolescents with a heart condition, and these include the even more pressing need to avoid ionising radiation, assessment of often complex anatomy that involves small structures, as well as imaging at a fast heart rate. Therefore, assessment for CHDs must involve a variety of modalities that can be used in a complementary fashion, and that together are sensitive, accurate, reproducible, and cost effective [36].

### 3.2. Imaging Modalities in Childhood

The imaging strategy for complex patients should be discussed in multidisciplinary meetings. The best risk-benefit imaging choice should consider age, clinical status, body size, radiation exposure and cardiac anatomy complexity.

Echocardiography remains the first-line imaging investigation for paediatric patients, as it is portable, non-invasive, and provides immediate, high-resolution anatomical and physiological information [37]. However, echocardiography fails when acoustic windows are poor, particularly for the assessment of extra-cardiac vascular structures.

Cardiac CT imaging is usually the best option in the context of poor acoustic windows when urgent imaging is needed, and prolonged general anaesthetic may carry a high risk.

The potential vascular complications as pre-assessment of pressure and oxygen saturation [38] and the dangers of exposure to radiation favour the use of CMR in patients [39] in whom haemodynamic data is essential (e.g., Fontan circulation or pulmonary hypertension) and interventional procedures are necessary. While cardiac catheterization was traditionally used to provide hemodynamic information and visualise extracardiac great vessels [40], CMR has progressively fulfilled this role [5].

CMR provides a powerful tool, giving anatomical and physiological information that echocardiography and catheterization alone cannot provide [41]. Extra-cardiac anatomy, including the great arteries and systemic and pulmonary veins, can be delineated with high spatial resolution. Vascular and valvular flow can be assessed, shunts can be quantified, and myocardial function can be measured accurately with high reproducibility, regardless of ventricular morphology [42].

Finally, CMR has proven to be better than both catheterization and echocardiography in providing high resolution imaging (without ionising radiation). In the paediatric population, CMR could be justified for any patient in whom clinical or echocardiographic data is insufficient for monitoring, decision making or surgical planning [43].

### 3.3. Indications

The decision to perform a CMR depends on the information required, the clinical state of the patient, and anaesthetic risks when this is required. CMR in children and adolescents with CHDs has a safety profile that needs to be considered. Clinicians together with radiologists should take into consideration the need of sedation, general anaesthesia or gadolinium contrast in order to balance risk.

An expert and trained anaesthetic team need to be involved with MRI safe equipment (blood pressure and heart rate monitor, electrocardiogram, pulse oximetry, pump infusion, expired gas concentration), and specific hemodynamic and imaging issues need to be discussed prior to each case. Prolonged and multiple breath-hold sequences are usually required and careful ventilation control is important to avoid hypoxia and hypercapnia [44]. This practise varies in different centres depending on local policy, but generally, children less than seven years of age or older with learning disabilities will have CMR performed under general anaesthesia. Other needed procedures can be carried out while the patient is under anaesthetia, as for example, estimation of pulmonary artery pressure through a needle transducer in the jugular vein in bidirectional Glenn patients before completion of total cavo-pulmonary circulation [45].

CMR requires significant expertise and training both in acquiring and interpreting the images. Patients’ body size, compliance, and rapid heart rate could represent technical and diagnostic challenges. CMR offers a comprehensive summary of the anatomical, post-surgical, and physiological status of the patient prior to being transferred to an adult service.

### 3.4. Sequences Used in Paediatric CMR

In contrast to scanning adults, the CMR in paediatric patients with CHD must be tailored and nearly all sequences need to be adapted to patient’s size, heart rate, and age and to respond specific clinical questions. Hence, the presence of both CMR and CHD experts is critical to optimize often long studies and to obtain high quality results [46,47]. Most common sequences are:**a.** **Spin-echo or “black-blood” sequences (BB SE):** blood appears dark whereas tissues are in shades of grey. These sequences are less susceptible to turbulent flow or metallic artefacts. Respiratory motion is controlled by breath-holding and multiple averages. Recommended for: anatomy definition, stents, complex CHDs [48,49].**b.** **Steady-state free precession sequences (SSFP):** blood appears bright. Two-dimensional (2D) SSFP is the most frequent sequence to assess anatomy, function, and valve motion in CHDs. SSFP cine slices stuck are usually used to assess ventricular volumes and ejection fraction. Cardiac motion controlled by retrospective ECG-gating. Recommended for: baseline heart assessment and CHDs including single ventricles [48]; see Figure 1.**c.** **Contrast-enhanced magnetic resonance angiography (CE-MRA):** gadolinium-based contrast administration increases the contrast between blood pool and surrounding tissue. Breath-holding and high spatial resolution sequences. No ECG-trigger, images reconstruction on average cardiac cycle. Recommended for: vessel anatomy assessment (aortic arch, pulmonary arteries and veins, collateral vessels) [50].**d.** **Three-dimensional (3D) SSFP sequences:** both ECG and respiratory gating (through diaphragmatic navigator) compensates respiratory motion. Contrast medium can be administered. High resolution 3D dataset of whole heart and intrathoracic vasculature. Recommended for: complex CHDs, proximal coronaries anatomy, visualisation of vessels anatomy. Limitation in case of low-pressure flow and stent [51,52,53].**e.** **Velocity-encoded phase-contrast (PC) cine sequences:** blood flow measurement across a vessel. Blood flows comparison is also fundamental to assess complex CHDs physiopathology. Recommended for: shunt assessment, collaterals, assessment of regurgitation or stenosis.**f.** **Late gadolinium enhancement (LGE) sequences:** abnormal deposition of contrast agent within the myocardium late after contrast medium injection. To detect myocardial fibrosis/scar (bright) in the contest of healthy myocardium (dark). Recommended for: cardiomyopathies, ischemic heart disease, myocarditis, CHDs (post- and pre-surgical scars) [48,54,55].

In our institution, the most common sequences performed during the MRI scan under general anaesthesia for patients with CHD are SSFP cines with mechanical breath-hold to assess anatomy, free breathing PC sequences for flow assessments, free breathing 3D SSFP (TSE, Turbo spin echo) sequences with respiratory motion compensation by diaphragmatic navigator for high-resolution whole heart dataset and, if needed, free breathing LGE sequences to detect myocardial fibrosis.

### 3.5. Clinical Applications of CMR in Childhood

**a.** **Aortic arch anomalies:** anatomical assessment for vascular rings, interrupted aortic arch, truncus arteriosus, aortic coarctation, connective tissue disorders (Marfan, Loeys-Dietz, Turner syndrome) [56]. Three-dimensional CE-MRA and SSFP sequences allow decision making for surgery or catheter-guided treatment [45]. Blood flow measurements with PC sequences provide information about vascular narrowing or presence of collaterals;**b.** **Pulmonary arteries:** CMR can provide detailed visualisation of spatial alignment of pulmonary artery bifurcation and side branches, as well as anatomical combined with functional information of aorto-pulmonary collaterals. Through-plane flow measurements can quantify flow distribution in between the lungs [57];**c.** **Pulmonary veins:** CMR is a fundamental tool to assess anomalous connection and stenosis of pulmonary veins. It can combine luminal anatomy (CE-MRA, 3D SSFP), accurate quantification of blood flow patterns (PC sequences) occurring in the presence of pulmonary venous obstruction and aorto-pulmonary collaterals [58];**d.** **Shunt lesions:** CMR is considered the gold standard for flow assessment. CMR can provide non-invasive anatomical detection of intracardiac and extracardiac shunts, flow direction and shunt quantification. Velocity-encoded PC sequences can accurately calculate Qp/Qs [59];**e.** **Tetralogy of fallot**: CMR enables assessment of right ventricular outflow tract (RVOT) and pulmonary arteries, as well as quantification of right ventricle volume and function and pulmonary valve regurgitation. Assessment of pulmonary flow distribution. Scar quantification. Key-imaging for surgical and/or percutaneous treatment decision making and follow up [60];**f.** **Complex CHDs:** CMR provide complete anatomical and haemodynamic information about situs, segmental cardiac connections, ventricular looping anomalies, intracardiac and extracardiac malformations, as tracheo-bronchial or abdominal anomalies and relationship [61,62]; see Figure 1;**g.** **Single ventricles:** CMR provides detailed anatomical, functional and flow assessment throughout palliative stages in single ventricle setting as well as during follow up after completion of the Fontan circulation (Fontan pathway obstruction, baffle leaks, lung flow distribution, thrombus formation and collateral flow) [63]; see Figure 2;**h.** **Cardiac tumours:** CMR represents a non-invasive instrument for tumour size and location, tissue characterisation, relationship and/or infiltration of surrounding structures, vascularisation and haemodynamic relevance [64];**i.** **Cardiomyopathies:** CMR provides non-invasive myocardial tissue characterisation (oedema, scar, replacement, and distribution of fibrosis) as well as global and wall motion abnormalities in idiopathic cardiomyopathy or secondary to neuromuscular disorders. Gold standard for follow up to measure response to treatments as well as family screening [65];**j.** **Myocarditis:** in the acute phase of myocarditis, CMR study is used to assess global ventricular function and regional wall motion abnormalities, early myocardial inflammatory changes (T2-weighted sequences), as well as myocardial necrosis/fibrosis (LGE sequences). In chronic myocarditis, CMR can be used to monitor biventricular function and to demonstrate inflammatory and fibrosis evolution. The LGE sequences typically visualise a patchy with a subepicardial and mid-wall contrast distribution most commonly detected in the lateral and inferior walls of the left ventricle. The presence of pericardial effusion provides supportive evidence for myocarditis [66].

### 3.6. Limitations

Despite CMR being a widespread imaging approach in childhood, it still has some technical limitations.

One of the challenges of CMR is that it is a long study, which can raise compliance problems for pediatric patients who may have difficulty remaining still for an extended period of time. However, the protocol can be tailored to include only the most essential images, reducing the length of the examination [67].

Another challenge of CMR in pediatric patients is the need for general anesthesia or sedation. Apart from the risk of anesthesia per se, which requires adequately trained staff, it is important to consider that general anesthesia may alter the physiological status of the patient and it can make it more difficult to achieve a good level of stress during a perfusion study, potentially leading to false negatives [68].

Device compatibility is another challenge of CMR. Some devices, such as pacemakers or implantable cardioverter-defibrillators (ICDs), are not MRI compatible. However, recent data in the adult population suggests that CMR may be safe for both MR conditional and non-conditional devices [69,70,71,72,73].

The use of contrast agents in CMR is another potential limitation. Gadolinium-based contrast agents (GBCA) have been associated with allergic reactions in a very small percentage of patients, with the majority of reactions being mild. Moreover, the use of GBCA in patients with severe kidney dysfunction (eGFR < 30 mL/min/1.73 m) has been associated with nephrogenic systemic fibrosis (NSF). Finally, accumulation of gadolinium in the basal ganglia of the brain has been reported in patients undergoing multiple administrations of GBCA in short time frames [74].

Inizio modulo

Fine modulo

## 4. Role of Cardiovascular Magnetic Resonance in the Adult Population

Technological progress and the robust evidence published in the literature, along with the unique advantages of the modality have contributed to expand and establish the role of CMR in several clinical contexts in the adult population [1].

In adults CMRs, the same type of sequences mentioned for the paediatric CMRs are used; in the following subparagraph, other specific sequences disease-based will be mentioned.

### 4.1. Congenital Heart Disease

As already described for the pediatric population, CMR is also key for the evaluation of adults with acquired cardiac diseases, cardiomyopathy, and repaired or unrepaired congenital heart disease, due to the unrestricted anatomical view provided and the possibility to accurately assessment volumes, flows, presence and extent of collaterals, thoracic and abdominal aorta. The absence of ionising radiation is an undoubtful advantage in these patients, often requiring serial assessment throughout their life. The accurate quantification of cardiac output and pulmonary to systemic flow (Qp:Qs) is often required to assess intracardiac shunts, guiding their therapeutical management [75].

Four-dimensional flow CMR is a novel phase-contrast sequence allowing a more comprehensive assessment of the multidirectional blood flowing inside heart and great vessels [76]. The fourth dimension (4D) is achieved resolving for the dimension of time a three-dimensional cine phase-contrast acquisition, with three-directional velocity-encoding (3D + time = 4D) [76]. This technique overcomes some of the limitations of Doppler and 2D flow (including data derived from geometrical assumption), allowing flow quantification in all the directions of the volume imaged. Its role in CHD has grown exponentially in the past few years, with increased use not only in research but also as a clinical tool. In this context in fact, the use of 4D flow can overcome the need of a precise 2D flow planning that is responsible often for longer acquisition times, avoiding potential inaccuracies in velocity measurements with a retrospective analysis of a single 3D volumetric acquisition [77]. This allows a better understanding of the physiopathological mechanisms subtended to several abnormalities, providing useful information for the pre-procedural planning.

### 4.2. Coronary Artery Disease (CAD)

CMR has a well-established role in both acute and chronic coronary syndromes, allowing not only the assessment of regional wall motion abnormalities and systolic function, but also the visualisation of acute myocardial injury and oedema [59,78,79,80]. The latter, identified through T2-weighted and T2 mapping images, may be particularly helpful in identifying the culprit lesion of the acute ischemic event in patients with multivessel disease and, when combined with LGE, in quantifying the myocardium salvage after revascularization [81]. CMR is also the current state-of-the-art model for the identification of microvascular obstruction (MVO), hypointense (“darker”) areas within the acute infarction, associated with worse cardiovascular (CV) outcomes [82,83]. Additional prognostic information can be derived from LGE sequences; the presence of a transmural scar is in fact associated with the absence of functional recovery and increased CV risk at follow up [6]. CMR is helpful in identifying myocardial infarction complications such as pericarditis, myocardial rupture, pseudoaneurysm or aneurysm. The possibility of adding a stress protocol to the information about scar and viability make CMR an optimal one-stop-shop imaging modality for patients with suspected ischemic heart disease. This requires the use of a vasodilator (more commonly, either adenosine 140–210 μg/min for 2–4 min, dipyridamole: 0.142 μg/kg/min over 4 min or a single bolus of Regadenoson 0.4 mg) or an inotropic agent (dobutamine 40 μg/kg/min ± atropine: 0.25 mg to a maximal dose of 2 mg) [84] with gadolinium-based contrast agent (GBCA) injected at maximal vasodilation. First pass perfusion images are therefore acquired (typically 3 short-axis images: base, mid, apical), and reveal inducible myocardial perfusion defect as hypointense (“darker”) areas within the normally perfused myocardium [84]. More recently, a novel automated in-line perfusion mapping method has also been developed to quantify myocardial blood flow (MBF) and myocardial perfusion reserve (MPR), with good accuracy compared to invasive parameters [85].

The pivotal role of CMR in patients with suspected or known ischemic heart disease is testified by current European and American guidelines. The 2019 European Society of Cardiology (ESC) guidelines recommend CMR in class I in patients with intermediate risk of CAD or as a non-invasive functional test if coronary computed tomography (CTCA) shows CAD of uncertain significance [7]. In addition, functional tests are recommended as the first approach to evaluate patients with high clinical likelihood of obstructive CAD, according to local expertise and availability. Most recent evidence prompted the recommendation in Class I in the 2021 American Heart Association (AHA)/American College of Cardiology (ACC) Guidelines for the evaluation of patients with acute and chronic chest pain and intermediate risk of CAD [86].

Several studies have been conducted to date to compare diagnostic accuracy among non-invasive imaging modalities, demonstrating higher sensitivity and diagnostic accuracy in CAD detection of CMR compared to single-photon emission computed tomography (SPECT) [87,88].

Approximately half of the patients presenting with anginal symptoms have no evidence of obstructive CAD at coronary angiography [89]. Quantitative stress CMR is particularly helpful in these patients with ischemia and unobstructed coronary arteries (INOCA), providing non-invasive assessment of microvascular dysfunction with good accuracy compared to invasively derived parameters [90,91]. In addition, between 5 and 15% patients presenting with acute coronary syndrome have non-obstructive CAD at invasive angiography [92]. Additionally, in this context, CMR plays a key role and both ESC and AHA/ACC guidelines recommend its use in Class I level of evidence B to identify the underlying etiology of myocardial infarction and unobstructed coronary arteries (MINOCA) and to distinguish true infarction from its mimics (takotsubo or myocarditis) [93,94]. The diagnostic yield is particularly high (74%) when performed within 14 days from the acute event, particularly when coupled with high (>211 ng/mL) troponin levels (94%) [95,96]; see Figure 3.

### 4.3. Cardiomyopathies

The presence and pattern of distribution of fibrosis detected by CMR is particularly helpful in the assessment of patients with known or suspected cardiomyopathy. Despite the fact that abnormalities in volumes and function can be easily picked up by echocardiography, an etiological diagnosis often requires further assessment with CMR and the identification of the distinctive pattern of fibrosis. The presence of mid-wall LGE characterizes approximately 1/4 of dilated cardiomyopathy (DCM) patients and is associated with an increased risk of death and arrhythmias [97]. In patients with hypertrophic cardiomyopathy (HCM) the fibrosis is often localized at RV insertion points and in the hypertrophied regions and correlate with a worse prognosis [98,99]. In these patients, CMR also allows accurate assessment of wall thickness and quantification of LV mass, along with the assessment of the left ventricular outflow tract (LVOT) obstruction using peak velocity measured in the in-plane flow images. In amyloidosis, CMR reveals some of the hallmarks of the disease (concentric LV hypertrophy, atrial dilatation, thickening of the valves), along with markedly increased native myocardial T1 and extracellular volumes (ECV) values and very fast uptake and clearance of GBCA with difficulties in nulling the myocardium [100,101]. In case of suspected myocardial iron deposition, CMR has a unique role in defining high risk patients (myocardial T2* < 10 ms) [102] that may benefit from iron chelating agents and in sarcoidosis allows the identification of myocardial infiltration or inflammation (increased signal in T2-weighted images) [103]. The pattern of LGE in this case may vary substantially, with virtually all patterns described in the literature. The Lake Louise criteria have been developed to provide a complete set of CMR diagnostic criteria to assist the diagnosis of myocarditis, using T1- and T2-weighted imaging, parametric mapping and supporting criteria as pericardial effusion and LV wall motion abnormalities [104]. The LGE pattern can be mid-wall or subepicardial and is thus particularly useful to rule out ischemic etiology in the acute setting, whose pattern would be typically subendocardial or transmural [94]. CMR is the gold standard for RV volumes and function assessment, part of the diagnostic criteria for arrhythmogenic cardiomyopathy [105]. In addition, the presence of fibro-fatty replacement, trabecular hypertrophy and of the “accordion sign” (multiple areas of outpouching with associated trabeculation, giving a crinkling appearance to the myocardium) may support the diagnosis, although it is not strictly part of the diagnostic criteria. Due to the unique properties already described, CMR is also the modality of choice for serial assessment of patients with suspected cancer treatment-related cardiomyopathy [106] or for the follow-up of patients with previous cardiac transplantation to assess long-term complications. Finally, apical hypertrophy (with or without apical thrombus) and associated diffuse subendocardial fibrosis, extending beyond a recognizable coronary territory, are typical of endomyocardial fibrosis [107].

### 4.4. Pericardial Disease

CMR allows detailed description of the pericardium, its thickness (normal when <3 mm), its relation with the surrounding structures and the presence of oedema and or fibrosis [1,72]. Fat saturation sequences are helpful to discriminate inflammation from epicardial or pericardial fat. Pericardial effusion can be visualised and quantified in bSSFP cine sequences with simultaneous assessment of its hemodynamic significance. Finally, in the suspicion of constrictive pericarditis, free-breathing real-time cine images can be acquired to assess the presence of ventricular coupling [108].

### 4.5. Cardiac Masses

A combination of cine images, T1- and T2-weighted, fat saturation, first pass, EGE and LGE sequences can be used to assess, localize and characterize cardiac masses [1,84]. Fat, recent hemorrhages and ferritin have in fact high signal intensity on T1-weighted images. Cysts have instead typically low T1 and high T2 signal. Increased signal in T2-weighted sequences and marked enhancement during first pass of GBCA are typical of highly vascularized lesions. Cardiac fibroma is instead typically hypointense in T2-weighted sequences. The pattern of LGE also allows for further distinction between malignant (often presenting a central, necrotic, hypointense area at EGE with heterogeneous LGE) and benign lesions (usually more homogenously enhanced at LGE). An inversion time (TI) set at approximately 500–550 ms at 1.5 T is optimal to null the signal deriving from a thrombus and to therefore differentiate it from other lesions [84].

### 4.6. Valvular Heart Disease and Great Vessels Assessment

Despite the fact that echocardiography remains the modality of choice for the evaluation of valvular heart lesions, CMR can be complementary, particularly in the case of poor acoustic windows. In addition, phase contrast images allow quantification of regurgitant volumes and transvalvular velocities without the geometrical limitation of Doppler assessment. Attention should be paid to adapt velocity encoding to the lowest value without aliasing and to use the lowest echo time (TE) when dealing with high flow jet velocities. Contrast-enhanced (CE) magnetic resonance angiography (MRA) is used to visualise lumen vessel with high spatial resolution, and can be used for a radiation-free assessment of the aorta or other vessels (i.e., renal arteries in the suspicion of secondary causes of hypertension). A contrast-free lumen visualisation is also feasible through other MRA techniques (i.e., “Fresh Blood Imaging”, 3D bSSFP or Quiescent Interval slice selective (QISS) MRA) and may be particularly useful in young patients or patients undergoing multiple serial assessments [87]. Next, 4D Flow CMR is a novel phase-contrast sequence allowing a more comprehensive assessment of the multidirectional blood flowing inside heart and great vessels [101]. The fourth dimension (4D) is achieved resolving for the dimension of time a three-dimensional cine phase-contrast acquisition, with three-directional velocity encoding (3D + time = 4D) [109,110]. This technique overcomes some of the limitations of Doppler and 2D flow (including data derived from geometrical assumption), allowing flow quantification in all the directions of the volume imaged. Although currently used mainly for research purposes, recent advances suggest their potential usefulness in aortic and valvular heart disease, as well as CHD being able to provide not only conventional flow parameters (such as mean or peak velocity or stroke volumes) but also additional fluid dynamic parameters (i.e., kinetic energy) that may help to better understand pathophysiology of several cardiovascular processes.

### 4.7. CMR during Pregnancy and Breastfeeding

There is often clinical concern when referring pregnant women to advanced imaging modalities. However, CMR is safe during pregnancy after the first trimester. Given the absence of radiation exposure, it is an attractive modality of choice to further assess pregnant patients with unconclusive echocardiographic examination, poor acoustic window [111,112] or for thoracic aorta assessment. A left lateral decubitus position may be preferred to avoid discomfort, particularly in the third trimester. Although not contraindicated during pregnancy, GBCA should be used only when it is expected to significantly improve fetal or maternal outcomes, given the low risk of stillbirth described, as well as neonatal death, and rheumatological and inflammatory disorders [113]. Although there is very limited GBCA excretion into breastmilk (<0.04% within the first 24 h) [114,115], in clinical practice it is recommended not to breastfeed 24h after GBCA administration.

## 5. Conclusions

Cardiac MRI offers a comprehensive and accurate cardiac assessment which can be performed safely and is highly reproducible throughout the different stages of fetal to adult life. When undertaken, CMR provides a detailed anatomical and functional evaluation at the different stages of life in a patient with cardiac disease and combines the ability to evaluate complex intracardiac and extracardiac anatomy with an accurate assessment of function, flow haemodyanmics and tissue characterisation, as well as providing a thorough assessment of progress or the response to cardiac surgery or cardiac interventions.

Additionally, three-dimensional (3D) reconstruction in CMR allows for a more comprehensive view of the heart’s structure and function, making it easier for clinicians to identify any abnormalities or changes that may be present. 3D reconstruction can provide better visualisation of complex structures, supporting clinicians to plan surgical or interventional procedures.

CMR minimises risks to the patient, particularly the apparent risks associated with more invasive haemodynamic assessments or from imaging modalities requiring exposure to ionising radiation such as cardiac catheterisation and cardiac CT. CMR as a modality is well established as the gold standard in chamber quantification and ventricular function assessment, with an extensive evidence base for use in the diagnosis and evaluation of a variety of cardiac pathologies in adults. As CMR technologies continue to develop, the image quality and breadth of data provided from this comprehensive assessment tool, when utilised independently or in combination with other more affordable and easily accessible imaging modalities, will continue to improve the diagnostic and therapeutic management of patients with cardiac disease.

## Figures and Tables

**Figure 1 children-10-00763-f001:**
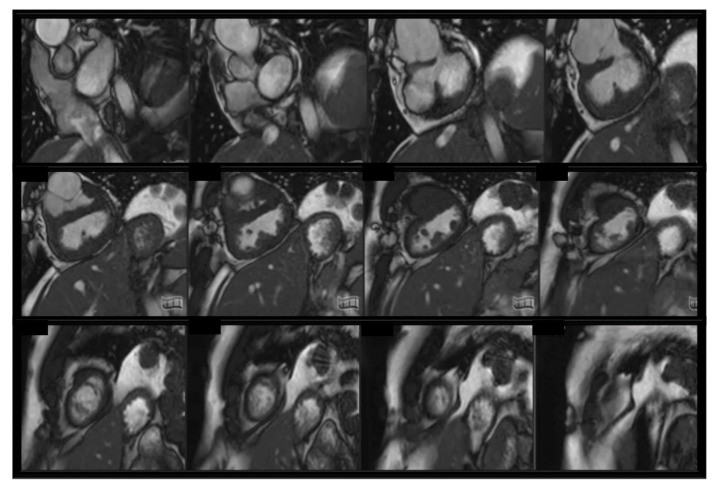
bSFPP cine stack in a complex CHD with univentricular physiology. We can observe biventricular hypertrophy, dilated aortic root, subaortic stenosis, inlet ventricular septal defect.

**Figure 2 children-10-00763-f002:**
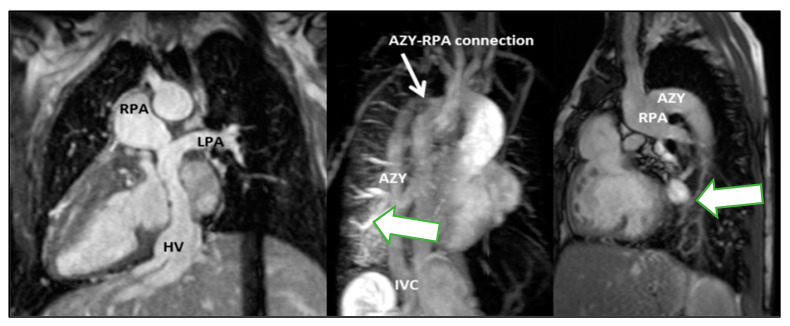
Total cavopulmonary connection (TCPC) is unobstructed (right pulmonary artery-RPA, left pulmonary artery-LPA, azygous vein AZY, hepatic veins HV). Pulmonary arteriovenous malformations in the right lower lobe, presumably due to streaming of hepatic venous flow to left lung (green arrow).

**Figure 3 children-10-00763-f003:**
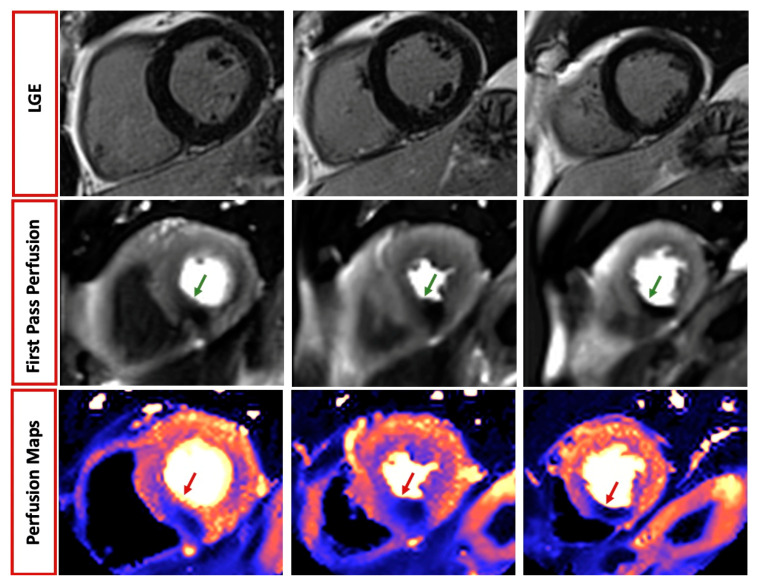
Role of cardiovascular magnetic resonance in patients with chest pain. A 59-year-old man presenting with angina and normal echocardiographic examination. Late gadolinium enhancement (LGE) sequences revealed absence of myocardial infarction or fibrosis. However, both qualitative and quantitative perfusion mapping demonstrated inducible perfusion defect during adenosine infusion (green and red arrow, respectively), suggesting significant underlying disease in the right coronary artery territory.

## Data Availability

Not applicable.

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
