# Peer review of "Cardiovascular Magnetic Resonance from Fetal to Adult Life—Indications and Challenges: A State-of-the-Art Review"

_children, 2023, doi:10.3390/children10050763_

Round 1
Reviewer 1 Report
Thank you for this great review paper on use of CMR. It is a very encompassing paper. I think it is well written but do have a few comments of things that need to be corrected.
1. You use the word foetal primarily but occasionally use fetal. You should be consistent throughout the paper.
2. Line 166 use of bSSFP sequences (this should have the full description prior to use of abbreviation. You describe this abbreviation later in the paper but should use full name with first use.
3. Line 200 ofcine should be of cine.
4. Line 230 should say Thus far instead of Still.
5. The Role of cardiovascular Magnetic Resonance during Childhood should include a comment about myocarditis.
6. Line 265 about non-sedated feed and wrap is not necessary for this paper.
7. In discussion about CMR being a powerful tool, there should be some comments about the potential of 3d reconstruction.
8. In the limitations of CMR, comment should be made for intracardiac devices such as pacemaker/defibrillator and stents.
Author Response
Dear Reviewer,
Thank very much for your comments
1. You use the word foetal primarily but occasionally use fetal. You should be consistent throughout the paper.
We have changed all the words in fetal.
2. Line 166 use of bSSFP sequences (this should have the full description prior to use of abbreviation. You describe this abbreviation later in the paper but should use full name with first use.
We have explained the acronyms in this line.
3. Line 200 ofcine should be of cine.
Corrected
4. Line 230 should say Thus far instead of Still.
Changed
5. The Role of cardiovascular Magnetic Resonance during Childhood should include a comment about myocarditis.
We have included a comment on myocarditis.
6. Line 265 about non-sedated feed and wrap is not necessary for this paper.
We took it off.
7. In discussion about CMR being a powerful tool, there should be some comments about the potential of 3d reconstruction.
We added a comment in the conclusion.
8. In the limitations of CMR, comment should be made for intracardiac devices such as pacemaker/defibrillator and stents.
We added a section on contraindications for CMR naming MRI devices non compatible.

Reviewer 2 Report
It is undoubtedly a high-level work by a group that performs cardiac resonance on a daily basis and, as far as can be seen, a large part of it is dedicated to patients with congenital pathology.
I would only suggest two things:
- Taking advantage of your experience, I think it would be of interest to provide a little more information regarding the way of acquisition of cine functional sequences in anesthetized pediatric patients. Do you perform free breathing cine with respiratory gating (according to which machine this option is only possible with SGRE cine, while the option for SSFP is not found), do you use real time sequences? In short, taking advantage of the fact that this is an article that places special emphasis on CMR in congenital patients, I think it would give even more value to the work than it already has, a section of small "tips and tricks" for pediatric patients with high heart rates and above all patients in whom the study has to be performed under general anesthesia, shedding light on how the group does it on a day-to-day basis.
- Finally, I am surprised at how little reference is made to 4DFlow. I only see mention of its use in the final section on valvulopathies. Our group performs 4DFlow in approximately 60% of the MRI studies in daily clinical practice, and we are not a group that has anywhere near the volume of congenital pathology that your group seems to have. I think that without a doubt the big star of 4DFlow is in the congenital pathology setting (not only for valvulopathies), where it helps enormously in a way that we would never have imagined in the early days of the technique. For those of us who have less experience in congenital pathology, it is a technique that is easy to acquire and enormously advantageous compared to two-dimensional sequences, which sheds light on these scenarios. I believe that there should be space for a section dedicated to 4DFlow in patients with congenital pathologies and reference some of the many articles that are coming out on the subject. The word "research use" is becoming a thing of the past for this technique. I am sorry, but I am totally in love with 4DFlow.
Otherwise, I can only congratulate the authors for their work.
- Two minor typing errors in "3 Foetal CMR applications" section:
Firs paragraph = Studies (insted of "tudies"
Second paragraph = of cine (instead of "ofcine")
The rest is perfect.
Author Response
Thanks very much for your comments. We really appreciate your inputs. Please find the answers below.
I would only suggest two things:
- Taking advantage of your experience, I think it would be of interest to provide a little more information regarding the way of acquisition of cine functional sequences in anesthetized pediatric patients. Do you perform free breathing cine with respiratory gating (according to which machine this option is only possible with SGRE cine, while the option for SSFP is not found), do you use real time sequences? In short, taking advantage of the fact that this is an article that places special emphasis on CMR in congenital patients, I think it would give even more value to the work than it already has, a section of small "tips and tricks" for pediatric patients with high heart rates and above all patients in whom the study has to be performed under general anesthesia, shedding light on how the group does it on a day-to-day basis.
We have written a brief summary of sequences usually used.
- Finally, I am surprised at how little reference is made to 4DFlow. I only see mention of its use in the final section on valvulopathies. Our group performs 4DFlow in approximately 60% of the MRI studies in daily clinical practice, and we are not a group that has anywhere near the volume of congenital pathology that your group seems to have. I think that without a doubt the big star of 4DFlow is in the congenital pathology setting (not only for valvulopathies), where it helps enormously in a way that we would never have imagined in the early days of the technique. For those of us who have less experience in congenital pathology, it is a technique that is easy to acquire and enormously advantageous compared to two-dimensional sequences, which sheds light on these scenarios. I believe that there should be space for a section dedicated to 4DFlow in patients with congenital pathologies and reference some of the many articles that are coming out on the subject. The word "research use" is becoming a thing of the past for this technique. I am sorry, but I am totally in love with 4DFlow.
Thanks for this great comment and for sharing this brilliant experience. We have added a paragraph for 4D flow.
Otherwise, I can only congratulate the authors for their work.
Comments on the Quality of English Language
- Two minor typing errors in "3 Foetal CMR applications" section:
Firs paragraph = Studies (insted of "tudies"
Second paragraph = of cine (instead of "ofcine")
The rest is perfect.
- We have revised the language accordingly.
